# “To Protect Myself, My Friends, Family, Workmates and Patients …and to Play My Part”: COVID-19 Vaccination Perceptions among Health and Aged Care Workers in New South Wales, Australia

**DOI:** 10.3390/ijerph18178954

**Published:** 2021-08-25

**Authors:** Katarzyna T. Bolsewicz, Maryke S. Steffens, Bianca Bullivant, Catherine King, Frank Beard

**Affiliations:** 1National Centre for Immunisation Research and Surveillance, Kids Research, Sydney Children’s Hospitals Network, Cnr Hawkesbury Rd & Hainsworth St., Westmead, NSW 2145, Australia; Maryke.Steffens@health.nsw.gov.au (M.S.S.); Bianca.Bullivant@health.nsw.gov.au (B.B.); Catherine.King@health.nsw.gov.au (C.K.); Frank.Beard@health.nsw.gov.au (F.B.); 2Hunter New England Local Health District, Health Protection, Wallsend Hospital Campus, Wallsend, NSW 2287, Australia; 3The Children’s Hospital at Westmead Clinical School, The Faculty of Medicine and Health, Cnr Hawkesbury Rd & Hainsworth St., Westmead, NSW 2145, Australia; 4School of Public Health, The University of Sydney, Edward Ford Building (A27) Fisher Road, Sidney, NSW 2006, Australia

**Keywords:** COVID-19, health workers, aged care workers, vaccination perception, vaccination acceptance, social and behavioural, risk communication, Australia, qualitative

## Abstract

The 2019 coronavirus disease (COVID-19) pandemic has globally caused widespread disruption, morbidity and mortality. The uptake of COVID-19 vaccination is critical for minimising further impacts of the pandemic. Health and aged care workers (HACWs) play a central role in public confidence in vaccines and are one of the priority groups for COVID-19 vaccination in Australia. Qualitative phone interviews with 19 HACWs aged 21–50 years old from New South Wales, Australia, were conducted, and the data were analysed thematically in order to understand the factors influencing HACWs’ acceptance of COVID-19 vaccination. We found that HACWs reported a continuum of COVID-19 vaccination intentions with 12 enthusiastically accepting and 7 hesitant. Using the Behavioral and Social Drivers of COVID-19 Vaccination (BeSD) Framework, we found that participants’ acceptance of vaccination was primarily driven by their perceptions of COVID-19 vaccination (such as safety, risk and benefits) and by the information sources, people and norms they trusted. Informed by study findings, we propose several communication strategies which may be helpful in addressing HACWs vaccination acceptance. We note however that as the pandemic continues, further studies with HACWs from diverse backgrounds are needed in order to provide accurate data on diverse motivational and practical drivers of evolving perceptions and attitudes towards vaccination.

## 1. Introduction

The 2019 coronavirus disease (COVID-19) pandemic, caused by the SARS-CoV-2 virus, has caused widespread disruption, morbidity and mortality globally. The uptake of COVID-19 vaccination will be critical for minimising further impacts of the pandemic [1]. The availability of vaccination, however, does not automatically translate into uptake. Vaccination uptake is influenced by various factors, including access to and affordability of immunisation services, awareness of vaccination, social norms, misinformation, perceptions of vaccines and attitudes towards vaccination [2,3].

Globally, including Australia, health and aged care workers (HACWs) have been prioritised to receive the COVID-19 vaccine due to high occupational exposure risk and increased likelihood of transmitting the virus to vulnerable people [4]. HACWs play a central role in the public confidence in vaccines [5,6]; their perceptions about the COVID-19 vaccination may influence not only their own but also the public’s vaccination decisions. Studies on the perceptions of HACWs, other adults in Australia and globally in relation to COVID-19 vaccination highlight various factors that may influence acceptance [7,8,9]; however, local and context specific evidence is needed to inform policy and practice.

This paper presents findings from qualitative interviews with HACWs in New South Wales (NSW), the most populous of the eight Australian states and territories. These interviews were conducted as part of a larger study aiming to identify factors influencing COVID-19 vaccination acceptance in three key groups (HACWs, older adults and adults with comorbidities) in order to inform risk communication messages.

## 2. Materials and Methods

Participants were recruited by a professional recruitment agency to meet predefined recruitment criteria, which included working as one of the following: doctor or nurse in hospitals, general practice or aged care, paramedic, pharmacist or non-medical staff providing face-to-face patient care in aged care settings in NSW. The participants received a reimbursement of AUD50 (USD37) for their time. The participants were sampled to ensure diversity relative to healthcare roles, professional settings and locations (major cities and inner and outer regional areas of NSW [10]). Those who provided written consent were scheduled for a 30 min phone or videoconference-based interview at a time that was convenient for them. The participants were asked about their perceptions of the COVID-19 pandemic, experiences with previous vaccinations, feelings about COVID-19 vaccination, intentions to vaccinate against COVID-19 and whether they would like to obtain further information to help them with their vaccination decision. Interviews were conducted between late January and early February 2021 prior to the COVID-19 vaccine rollout to HACWs in Australia, which commenced in late February 2021.

The interviews were conducted by MS, BB and KB and analysed together with CK, all of whom are experienced public health researchers skilled in qualitative research methods. Interviews were audio-recorded and professionally transcribed. Thematic analysis involved reading and re-reading transcripts, highlighting and extracting information relevant to the key research question [11]. A pragmatic lens guided the analysis, aiming to produce findings that could be translated into risk communication messages in order to support vaccination acceptance among HACWs.

A criteria of rigour relevant to qualitative methods were applied to the study design and data analysis [12]. In order to ensure reliability and validity of interpretations, we comprehensively analysed all transcripts using an iterative and collaborative approach, compared findings across cases and provided evidence from the data on interpretations made [13,14]. Each transcript was read separately by two researchers who then created a combined analysis file. The team met weekly to discuss combined analyses, reconcile differences in interpretation and agree on the emerging categories. Informed by the analysis of an initial eight transcripts, KB drafted a framework of key categories and sub-categories. The framework was used for the remaining analysis, during which it was expanded with new insights and refined. From that refined framework, overarching themes were identified in a group process. Findings were loosely mapped against the World Health Organization Behavioural and Social Drivers (BeSD) of COVID-19 Vaccination Framework [2], which is based on the ‘increasing vaccination model’ [15]. The BeSD Framework groups factors influencing vaccination behaviour into either motivational factors (influenced by what people think and feel about vaccination and social processes) or practical issues surrounding access to vaccination. The BeSD guidebook suggests that, depending on the types of factors found in a given context, different types of public health interventions should be employed [2].

The study received approval from the University of Sydney Human Research Ethics Committee, 2020/277.

## 3. Results

### 3.1. Participants

In total, 19 participants aged between 21 and 50 years were interviewed (Table 1).

### 3.2. Findings

Three following themes were identified.

**Theme** **1.**
*There is a continuum of COVID-19 vaccination intentions among HACWs.*


Participants reported a range of COVID-19 vaccination intentions, from enthusiastic acceptance (12 participants), accepting but reporting no choice in the vaccination decision due to professional role (3 participants), hesitant (i.e., wanting to “wait and see”, wanting to obtain further information before making a decision or having specific concerns) (3 participants) and rejecting (1 participant).


*“In terms of the vaccine itself, I would say I’m very positive and very excited about it. I can’t wait to get my own! (enthusiastically accepting).”*



*“If it was mandatory I guess I kind of have to get it. (accepting but reporting no choice).”*



*“I guess it’s because I think with cases being relatively low in Australia and my risk of actually contracting the disease and I guess from just a selfish perspective, if I don’t need it and I can give it time and see how it pans out a bit once it’s pushed to the public. (hesitant, wanting to ‘wait and see’).”*



*“I’m not willing to get the vaccine any time soon unless I become really forced to do it. If I don’t get forced to do it, I’m not going to do it. (rejecting).”*


In this paper we refer to enthusiastic acceptors of COVID-19 vaccination as “accepting”. Consistent with the definition of vaccine hesitancy [15], we refer to participants who accepted but reported no choice due to professional role, participants who hesitated towards and one participant who rejected COVID-19 vaccination as “hesitant”.

**Theme** **2.**
*Various perceptions influence COVID-19 vaccination acceptance among HACWs.*


Participants discussed COVID-19 vaccination through a prism of different perceptions that had varying influences on vaccination acceptance (hereafter facilitators and barriers). We found six common perceptions (Table 2) that acted as either facilitators or barriers to acceptance.

#### 3.2.1. Perception 1: Level of Trust towards Science, Vaccine Development Process and Health Authorities

Accepting participants reported trust in science and research processes in general and, in particular, those underpinning COVID-19 vaccine development. Some expressed awe of the technology that was used to develop mRNA vaccines, describing it as a scientific breakthrough. Vaccine accepting participants were also more likely to report trusting health authorities and vaccine safety systems funded and operated by the government (for example, the Therapeutic Goods Administration (TGA) in Australia). They referred to the hard work being conducted “*behind the scenes*”. They trusted that the accelerated process of vaccine development did not compromise vaccine quality or safety. Many appreciated the transparency and timeliness with which they perceived the Federal and NSW State government shared COVID-19 information.


*“So, I guess I just trust the scientific process. And then if the people that dedicate their lives to trying to develop and test these vaccines are giving it the go ahead, as long as there’s nothing shifty going on behind the scenes in the government, I’m all for it.”*



*“I have faith in things like the TGA… like some of the major scientific bodies, like the Lancet …I have a lot of trust in scientific rigour of certain journals and studies.”*


Conversely, hesitant participants reported having many concerns about vaccine effectiveness and safety and questioned the speed of vaccine development. A few hesitant and/or refusing participants used language pitting themselves (“us”) against the government and pharmaceutical companies (“them”). Some in that group perceived that the information about COVID-19 and COVID-19 vaccines coming from ‘them’ had been scarce and not transparent. A few also reported misperceptions based on misinformation and conspiratorial thinking, for example, that information about serious adverse events following vaccination had been intentionally withheld.


*“the medical or the pharmaceutical companies, they’re like pushing it because they make billions and billions of dollars… It’s going to be advantageous for them, if everyone gets vaccinated and the government pushes that idea.”*



*“I read articles that just really shook my nerves. They said they’re using aborted babies–male aborted babies that are like that are up to six months old to obtain whatever RNA they needed. It’s like gee, that is horrible. And I thought there’s no way I will come near this vaccine.”*


The speed at which government information and recommendations about COVID-19 and vaccination changed was reported by some as contributing to a sense of uncertainty and mistrust towards official information. One participant said he “*lost*
*trust in a lot of information”*.
*“Because there is no consistency, it is like, you wonder, how true is this information?”*

#### 3.2.2. Perception 2: Level of Confidence in COVID-19 Vaccination

Most participants reported confidence in vaccination in general and used examples of how vaccinations have helped eradicate vaccine preventable diseases. Most participants also described positive experiences with routine vaccination and would regularly vaccinate against influenza.


*“I’m for vaccinations and I think for general public health measures, vaccinations seem to be proven to be a good and effective strategy for many different illnesses.”*


A small number of accepting participants also highlighted the effectiveness of COVID-19 vaccines in reducing hospitalisation.


*“… the AstraZeneca vaccination looks like… it decreases hospitalisation by almost 100%. So in terms of the public health measure to prevent hospital load of COVID patients, that’s obviously going to be extremely effective.”*


Hesitant participants, on the other hand, reported low levels of confidence in COVID-19 vaccination and reported many concerns, discussed below under ‘Risk/benefit calculation’.

#### 3.2.3. Perception 3: A Sense of Obligation to the Broader Community to Vaccinate

Accepting participants were more likely to describe wanting to vaccinate to protect others at work (patients and co-workers), at home (family and housemates) and in the community (friends and vulnerable community members). One person put a moral value on vaccination, perceiving vaccination as *“doing your bit”* and being “*a good citizen”.*


*“Anything that I can do to help the greater community, especially in, you know, going to lots of people who are susceptible to getting diseases, I think…-it’s not much work for me, and it will do- it will help them in the long run yeah.”*



*“To protect myself, my friends, family, and workmates and people that I work with, clients, patients, and to play–play my part.”*


This sense of obligation to the broader community to vaccinate was not reported by hesitant participants. In fact, some in that group questioned the need for the “young and healthy” (including themselves) to vaccinate at all. A few also reported confusion with the concept of herd immunity. This indicates that lack of knowledge is potentially contributing to some HACWs’ poor sense of community values of vaccination.


*“Well, I just hear them thrown around on the news, like if we have herd immunity does that mean everyone is vaccinated? Like immunity, it’s just the weak, the more vulnerable, the weaker ought to be vaccinated, sure, that would be great. But for a healthy person, you know, no medical history issues, I don’t believe we should force them.”*


#### 3.2.4. Perception 4: Perception of COVID-19 Disease Risk

Accepting participants were more likely to describe the COVID-19 pandemic as serious, with global reach and to stress the negative impacts it has had on people’s lives.

They also reported high work-related risk of exposure to COVID-19. This was most pronounced for those HACWs who had experienced COVID-19 first hand at work, for example, by conducting COVID-19 testing or seeing patients with COVID-19 disease. Many reported worrying about inadvertently passing on the virus (at work, at home and in the community) and the devastating consequences (serious disease and death) this could have on those that are most vulnerable.


*“… we went into a couple of nursing homes and I had a patient who was over the age of 90 and just thinking like if I was the one to give that person COVID and they were to pass away like that would just, I don’t know, that breaks my heart.”*



*“Yes, there needs to be more research. I get that. But if I gave COVID to someone and I killed them, essentially, or I passed it onto them and they died, I wouldn’t forgive myself.”*


While most HACWs did not feel at risk of serious disease (due to being young and healthy), a few participants reported a sense of heightened personal risk due to the underlying health conditions in themselves or in immediate family members.


*“I feel a sense of mild anxiety about getting COVID. No one really knows what the outcome of someone who is young, physically healthy, but immunosuppressed, would be. So I do think on these things a lot more.”*


Conversely, hesitant participants were more likely to report that the pandemic risk in Australia was overstated or politicised. Some proposed that COVID-19 may be similar in severity to seasonal influenza. This was particularly the case for those who reported not knowing anyone who has been infected with COVID-19. On the premise of “*not having COVID in the community at the moment”*, they questioned the rush to vaccinate.


*“It’s not a deadly virus. It’s not a poison.”*



*“We’ve had how many cases and they’re just cases, and they have to obviously self-isolate and so they’re kind of-if this COVID is just a new flu, like I don’t know.”*


#### 3.2.5. Perception 5: Risk/Benefit Calculation

Accepting participants were more likely to perceive that, while COVID-19 vaccines come with some risks, the benefits of vaccinating greatly outweighed those risks.


*“It hasn’t been trialled for years and years…but what we do know is the short term and some of the concerns regarding longer-term effects of not getting the vaccine.”*


Some saw COVID-19 vaccination as *“necessary*” and *“the best chance”* given that there was no cure or a single solution to the COVID-19 pandemic.


*“I would get the COVID-19 vaccine because I guess as a public health measure, it seems to be the best chance we have currently to turning cases around and improving general public health outcomes for everyone globally.”*


Some used combat metaphors to describe their perceptions of COVID-19 vaccination, such as referring to COVID-19 vaccination as a *“weapon”* in “*a war on this illness”.*

A large number of accepting participants discussed vaccination as a means of returning to *“normality”*, including unrestricted social interactions, events, domestic and international travel and entertainment.


*“We need normality back. We need to be able to walk around with no masks on, be able to hug people again, be able to go and visit our grandparents in nursing homes. Life needs to be normal again. It’s too much. We need to be able to travel again. We need to be able to have fun again. And I think that’s what the vaccine–the vaccine will bring normality.”*


Other reported benefits of COVID-19 vaccination included the following: protection against infection with the virus and against developing serious illness and an associated sense of personal safety; the ability to continue to work and prevent economic impacts; protection for family members living overseas; and an associated sense of relief that they would be safe once vaccinated.


*“I think, just from a risk benefit perspective, I think it’s got a low level of risk and the potential of benefit is that if there was another major outbreak, that I’d feel more protected against it, not only from stopping me getting sick but stopping me from getting… COVID, being asymptomatic and giving it to patients. So, I feel like it’s an extra safety measure on top of PPE.”*



*“I will definitely be getting vaccine- I will feel safer, so that I don’t catch it. And less afraid of going out.”*


Conversely, hesitant participants were less likely to describe the benefits of COVID-19 vaccines and more likely to voice concerns. Concerns (summarised in Table 3) could be organised into those about COVID-19 vaccine safety and effectiveness and those about the logistics of the vaccine rollout (i.e., when and where it will be available).


*“COVID’s so new, it’s from last year or the year before and then they’re making this really quickly and then they’re releasing it. What are the long-term side effects is definitely a concern and…a lot of people ask me that as well.”*



*“… we don’t have specific freezers, monitoring of freezers. So for me I feel like there could be some errors or mistakes made or the cold chain broken. So that for me is a little bit of concern.”*


#### 3.2.6. Perception 6: Perception of Personal Agency

Accepting participants reported feeling that they had a choice to vaccinate or not. They referred to making that choice to proactively take care of themselves and others. In that sense, COVID-19 vaccination was observed as a means to exercise personal agency around personal and public health.


*“I am protecting myself and I am protecting the people that I look after and my co-workers. I am protecting my partner against it, the family.”*


Conversely, hesitant participants discussed COVID-19 vaccination through a lens of limited choice and control. They used expressions such as *“we are being dictated to”* and *“we are being told to accept it as it is”.* One person stated that, while the vaccines were not explicitly mandatory, people who object would essentially be barred from having a *“normal life”* of travelling, working in a clinical capacity or being eligible for government support. Another person expressed fear and anger that he and other young and healthy people will be used against their will “for testing” of COVID-19 vaccines.


*“They do say oh, it’s not going to be forced, but if you don’t have the vaccine you’re not allowed to do any overseas travel … -so you’re a prisoner if you don’t.”*



*“I don’t want to be the first person to put my hand up and say here, you can use me for tests basically.”*


**Theme** **3.**
*Information sources, people and norms influence HACWs perceptions of COVID-19 vaccines.*


The participants discussed used and relied on various sources of COVID-19 vaccination information. Some accepting participants reported knowledge gained from their clinical training and/or professional experiences (i.e., being a vaccination provider and planning to be COVID-19 vaccination provider) as helpful in understanding various immunisation concepts (herd immunity; transmission of the virus to vulnerable people) and processes of vaccine development. They referred to this information, as well as that from health authorities (such as the NSW Health website), as “reliable”. This knowledge helped instill confidence in the vaccines, for example, that vaccine safety had not been compromised despite accelerated processes. Participants who reported knowledge gained from medical training and/or professional experience were more likely to critique mainstream and social media as an “unreliable” source of information.


*“I am getting most of my information from the Health website, and things like journals and that kind of thing. I think that comes from the health background and not Googling it.”*



*“I try and get my news and articles from reputable sources not tabloid news journalism… Something that’s evidence-based or backed by research, not so that they’re just opinion pieces.”*


Participants also reported accessing mainstream and social media, but some hesitant participants reported confusion with the COVID-19 vaccination content presented there. This content included the changing information and advice on the COVID-19 pandemic and misinformation about COVID-19 vaccination.


*“You can go online and a video will come up on your Facebook or your Instagram and it will be some lady having seizures and stuff, post-vaccination. But then you don’t know whether to believe that was actually the case of the seizures and whatnot.”*


## 4. Discussion

Most HACWs in this study were enthusiastic acceptors, followed by participants who accepted but reported no choice due to their professional role and hesitant participants. There was a single case of vaccine rejection. These variabilities and distributions are consistent with the continuum of vaccination positions documented in the literature [4,16]. The relative lack of strongly rejecting individuals is a positive finding for communication planning to strengthen vaccine acceptance among HACWs [17,18], although caution needs to be taken with interpreting results given the small sample size.

### 4.1. BeSD Framework and Key Findings

In this study, HACWs’ perceptions fell mostly under the BeSD motivational domains of “thinking and feeling” and “social processes” [2]. Only a small number of participants discussed practical issues such as vaccine supply or concerns about the cost of vaccination. This is understandable as, in line with the study focus, we did not probe about factors influencing vaccine uptake. Practical concerns may also not have been front of mind for participants at the time of the interviews, which occurred prior to vaccination rollout.

Our findings that accepting participants perceived their risk of developing COVID-19 disease as high; reported confidence in vaccine effectiveness and safety; and expressed trust in the scientific processes underpinning vaccine development, the government and health systems resonate with recent literature [6,19,20]. Our findings are also similar to those from an online survey conducted in a representative sample of the broader Australian population in August 2020, in which those who expressed confidence in governments and the health system were more likely to intend to be vaccinated [8]. Likewise, our findings that hesitant participants expressed concerns about vaccine safety, effectiveness and perceived scientific uncertainty; reported low perceived risk of developing severe COVID-19 disease; expressed doubts about the seriousness of the pandemic and lack of trust in authorities; and may have been influenced by COVID-19 misinformation and conspiracy theories are also corroborated by the recent literature [21,22,23,24].

Our study also highlights some barriers and facilitators to COVID-19 vaccination acceptance which, although recently identified in the literature [9,19,21], have received limited attention.

First, we found that accepting participants expressed a strong desire and sense of responsibility to receive a vaccine in order to protect vulnerable people, a finding corroborated by two recent studies [9,21]. In contrast, hesitant participants questioned the need for the entire community (including healthy people such as themselves) to be vaccinated rather than just those vulnerable to severe COVID-19 disease, such as the elderly or those living with comorbidities.

Second, accepting participants reported positive vaccination influences from trusted medical colleagues and reported having access to what they perceived as reliable, evidence-based information. Receiving a recommendation to vaccinate against COVID-19 from one’s health provider and/or a reputable scientific source was similarly reported as cue to vaccinate in two recent studies [9,19]. In our study, accepting participants critiqued the reliance on social and mainstream media for information about COVID-19 vaccines. Hesitant participants, on the other hand, reported negative social influences and concerns about the available information on COVID-19 vaccines. They reported encountering negative information about COVID-19 vaccines, especially on social media, and struggled to determine the credibility of this information.

### 4.2. Recommendations for Communication Strategies to Increase COVID-19 Vaccination Acceptance among HACWs

Recent papers recommend various strategies to address previously identified barriers of COVID-19 vaccination acceptance [25]. Here, we use the BeSD Framework and communications literature to recommend communication strategies for increasing COVID-19 vaccination acceptance among HACWs in light of our study’s two novel findings. If incorporated into policy and practice, these recommendations should help improve COVID-19 vaccination uptake among HACWs in Australia.

Firstly, doubts among hesitant individuals about the importance of whole-of-community vaccination, categorised in the BeSD Framework as a “thinking and feeling” factor contributing to vaccine uptake, are best addressed with educational campaigns and educational materials [2]. Accordingly, informed by our study’s findings, resources could be developed to explain the communal benefits of vaccinating, such as preventing outbreaks, enabling travel and reducing the social and economic impacts of the pandemic [26]. On the other hand, as evidenced by a recent study with 15,000 individuals in the UK, some individuals may not be sufficiently motivated by a better understanding of the communal benefits of COVID-19 vaccination [27]. In their vaccination conversations with patients, public health communicators and clinicians should, therefore, consider emphasising both the communal and individual benefits of vaccinating, with the latter including protecting oneself against the virus and from the individual-level health, economic and social impacts of the COVID-19 illness [26]. We further suggest that COVID-19 vaccination public communication and resources suitable for use by clinicians should be in easy-to-digest formats, for example, short videos, infographics and tables. Humour, both in the content and format of messages (e.g., short videos and cartoons), may also be considered and has been used effectively in the national COVID-19 messaging campaign in the UK [28].

Secondly, negative information and social influences, categorised as “social process” factors in the BeSD Framework, are best addressed by vaccine championing and strong recommendations to vaccinate from both institutions and trusted health providers [2]. Vaccine acceptance may be further enhanced by access to trusted, reliable sources of information about vaccination [5,25,29]. Accordingly, informed by our study’s findings, we recommend developing communication campaigns targeting HACWs that feature trusted medical peers who clearly recommend vaccinating against COVID-19 (politicians were perceived by some participants as less trustworthy sources of information and, therefore, should not be featured in such campaigns). Given the risk of evidence-based information about vaccines being distorted and misrepresented on social media [30,31,32], such campaigns should offer opportunities for two-way communication with HACWs (i.e., via online or public forums, telephone hotlines or social media) so that individuals are able to seek answers to any specific questions they may have. During two-way communication, any misinformation should be directly addressed by explicitly refuting it [33,34]. In order to debunk myths more effectively, detailed corrective information should be provided [33,34]. Clinicians could also be recruited as “COVID-19 vaccination champions” and be provided with training and/or resources to facilitate their engagement with colleagues for promoting vaccinations. We further recommend that additional trusted information channels (beyond government websites) that are particularly accessible to HACWs employed in the private sector should be investigated.

### 4.3. Future Research

While COVID-19 vaccination in Australia is offered to eligible individuals at no cost, there may be factors other than vaccine cost that hinder access for the public and HACWs, including awareness of services, booking systems, geographical distance and transport to services, service convenience and cultural appropriateness [3,35,36]. Further qualitative research with priority populations from different geographic, social and cultural environments conducted during different stages of COVID-19 vaccine rollout will provide insights into motivational and practical factors that influence vaccine acceptance and uptake over time.

Mixed methods research can also provide insights into different sources of vaccination information that groups perceive as trusted and reliable. It is important that such insights inform future interventions intended to support public acceptance and uptake of COVID-19 vaccines.

Some participants discussed their uncertainty with certain aspects of the rollout of COVID-19 vaccines and how this could limit their ability to discuss or recommend the vaccines to patients. Further research into the needs of immunisation providers for having supportive conversations with patients is needed. This could include providing immunisation providers with written resources or training modules.

Finally, many participants reported various negative impacts of COVID-19, including economic, social and mental impacts. Further qualitative research is needed in order to better understand COVID-19 impacts on HACWs’ lives.

### 4.4. Limitations

As this is a qualitative study, we were not concerned with external validity [13,37] and used rigorous methods instead to ensure that interpretations and findings reliably reflected participants’ perspectives. Experiences and views presented here may not represent those of HACWs from outside NSW and from diverse social and cultural backgrounds. With the focus of the research on COVID-19 vaccine acceptance, we have not explored practical issues that may influence COVID-19 vaccine uptake. Interviews were conducted immediately prior to the vaccination rollout in Australia; HACWs perceptions related to COVID-19 vaccination may have changed since. Australia has also had a very limited number of COVID-19 cases to date, which is different relative to the context in many other countries (i.e., hesitant participants in this study reported not having seen anyone with COVID-19).

Findings from the study, however, provide a rich description of perceptions from HACWs working across NSW about COVID-19 vaccination and factors influencing COVID-19 vaccination acceptance.

## 5. Conclusions

This qualitative study with HACWs in NSW, Australia, prior to vaccination rollout provides information on the continuum of vaccination positions among HACWs and factors influencing HACWs’ acceptance of COVID-19 vaccination. It confirms and expands on the findings from recent literature. New insights highlight the importance of publicly acknowledging that many HACWs make community-focused vaccination decisions; identifying and enabling HACWs’ access to reliable, evidence-based and trusted sources of vaccination information; making vaccination related information and concepts easy to understand; and actively debunking misinformation. Communication about COVID-19 vaccines that is based on an understanding of factors influencing COVID-19 vaccine acceptance can build public trust and more effectively support the acceptance of COVID-19 vaccines. This study provides such evidence and suggestions for the content of risk communication; thus, the findings have direct implications for clinical practice, public health policy and practice. With COVID-19 vaccine boosters potentially required at annual or other interval, future studies should explore factors that influence not only COVID-19 vaccination acceptance but also the uptake in priority populations from diverse geographic and social environments.

## Figures and Tables

**Table 1 ijerph-18-08954-t001:** Characteristics of study participants.

Category	Number of Participants
Workplace setting	
Healthcare (including registered nurses, doctors, paramedics and pharmacists working in hospitals, primary care and frontline emergency settings)	15
Aged care (including clinical staff (nurses) and non-clinical staff: aged care workers and exercise physiologist)	4
Gender	
Female	10
Male	9
Age group	
20–29 years old	9
30–39 years old	9
40–50 years old	1

**Table 2 ijerph-18-08954-t002:** Perceptions acting as either facilitators or barriers to COVID-19 vaccine acceptance.

Perception	COVID-19 Vaccine Accepting Participants Were More Likely to Report the Following	COVID-19 Vaccine Hesitant Participants Were More Likely to Report the Following
1. Level of trust towards science, vaccine development process and health authorities	Trust in the science and research processes underpinning COVID-19 vaccine development.	Distrust towards the scientific process underlying COVID-19 vaccine development.
2. Level of confidence in COVID-19 vaccination	Trust in COVID-19 vaccination effectiveness.	Low levels of confidence in COVID-19 vaccination; many concerns about vaccination.
3. A sense of obligation to the broader community to vaccinate	Wanting to vaccinate against COVID-19 to protect others; discussing moral imperative to vaccinate	Not understanding and/or questioning the need to vaccinate oneself to protect others.
4. Perception of COVID-19 disease risk	COVID-19 disease perceived as serious; feeling personally at risk; knowing someone infected with COVID-19.	COVID-19 pandemic perceived as overstated or politicised; not feeling personally at risk; not knowing anyone infected with COVID-19.
5. Risk/benefit calculation	Benefits of vaccinating greatly outweighed potential risks.	Many concerns about COVID-19 vaccines; benefits of vaccinating unclear.
6. Perception of personal agency	Expressing a choice on whether to vaccinate against COVID-19; vaccination seen as a proactive way to take care of self and to protect others.	Expressing concerns about not having a real choice on whether to vaccinate against COVID-19; COVID-19 vaccination seen as enforced and unwelcomed.

**Table 3 ijerph-18-08954-t003:** Summary of concerns about COVID-19 vaccination.

Group of Concerns	Examples
Concerns about COVID-19 vaccine safety and effectiveness	What is in the vaccines Newness of mRNA technology Safety Efficacy Effectiveness ○ Against new variants ○ Longevity of the immune response to COVID-19 vaccine Side effects ○ Short-term ○ Long-term, for example, potential impacts on fertility and/or health of unborn children
Concerns about logistics of vaccine rollout	Potential for a break in cold chain Storage for vaccines to meet the demand When and where vaccines will be available

## Data Availability

Data collected and used in the analyses are not publicly available.

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
