# Peer review of "“To Protect Myself, My Friends, Family, Workmates and Patients …and to Play My Part”: COVID-19 Vaccination Perceptions among Health and Aged Care Workers in New South Wales, Australia"

_ijerph, 2021, doi:10.3390/ijerph18178954_

Round 1
Reviewer 1 Report
Thank you for your interesting work. A few questions for your consideration are:
- What are the limitations related to validity and reliability of this work?
- What are the implications for clinical practice, future research, and public health policy or promotion?
- There are studies on healthcare workers perception of COVID-19 vaccine and their hesitancy, many of these studies were not cited and how closely are your results aligned with these:
https://link.springer.com/article/10.1007/s10900-021-00984-3
https://www.sciencedirect.com/science/article/pii/S2666354621000922
https://onlinelibrary.wiley.com/doi/10.1111/jep.13581
Author Response
Dear Reviewer 1 and the Editor
Thank you for your review and feedback. Please see the attachment.
Kind regards

Reviewer 2 Report
The paper submitted for review is an interesting survey of the perception of the need for vaccination against Covid-19 by health and social care workers in Australia just before the start of the vaccination program in the general population. The paper is original, well-constructed, and well presented, and can be of particular interest as the vaccination campaigns in most countries are still ongoing, containing some reflections that may allow adjusting the focus in convincing as many people as possible to adhere to the vaccination. The study was set up as a science-based experimental project, but in reality, it ends up being only a descriptive representation due to the small number of people involved even if representative and qualified. It would have been very interesting to place side-by-side data from this group of health professionals an analogous number of opinions coming from uniform strata of the general population. In the presented results, it is striking that as many as 30% of these health workers, who could probably be personally involved in the vaccination campaign, were hesitant about the COVID-19 vaccination. The resulting message should make us reflect on the difficulty that data from official science are perceived correctly by the world of social media, which greatly influences the general population. The main comments on the article are to try to shorten the description of the results and to avoid the restatement of the same in the discussion that is a bit lengthy
Author Response
Dear Reviewer 2 and the Editor
Thank you for your review and feedback. Please see the attachment.
Kind regards

Round 2
Reviewer 1 Report
Thank you for the revisions